# Long-Horizon Visual Planning
# with Goal-Conditioned Hierarchical Predictors

**Karl Pertsch**[*,1]   **Oleh Rybkin**[*,2]   **Frederik Ebert**[3]
**Chelsea Finn**[4]   **Dinesh Jayaraman**[2]   **Sergey Levine**[3]

[1] USC   [2] UPenn   [3] UC Berkeley   [4] Stanford University

## Abstract

The ability to predict and plan into the future is fundamental for agents acting in the world. To reach a faraway goal, we predict trajectories at multiple timescales, first devising a coarse plan towards the goal and then gradually filling in details. In contrast, current learning approaches for visual prediction and planning fail on long-horizon tasks as they generate predictions (1) without considering goal information, and (2) at the finest temporal resolution, one step at a time. In this work we propose a framework for visual prediction and planning that is able to overcome both of these limitations. First, we formulate the problem of predicting *towards a goal* and propose the corresponding class of latent space goal-conditioned predictors (GCPs). GCPs significantly improve planning efficiency by constraining the search space to only those trajectories that reach the goal. Further, we show how GCPs can be naturally formulated as hierarchical models that, given two observations, predict an observation between them, and by recursively subdividing each part of the trajectory generate complete sequences. This divide-and-conquer strategy is effective at long-term prediction, and enables us to design an effective hierarchical planning algorithm that optimizes trajectories in a coarse-to-fine manner. We show that by using both goal-conditioning and hierarchical prediction, GCPs enable us to solve visual planning tasks with much longer horizon than previously possible.

## 1 Introduction

Intelligent agents aiming to solve long-horizon tasks reason about the future, make predictions, and plan accordingly. Several recent approaches [11, 71, 19, 69, 42, 21] employ powerful predictive models [16, 4, 20, 37] to enable agents to predict and plan in complex environments directly from visual sensory observations, without needing to engineer a state estimator. To plan a sequence of actions, these approaches usually use the predictive model to generate candidate roll-outs starting from the current state and then search for the sequence that best reaches the goal using a cost function (see Fig. 1, left). However, such

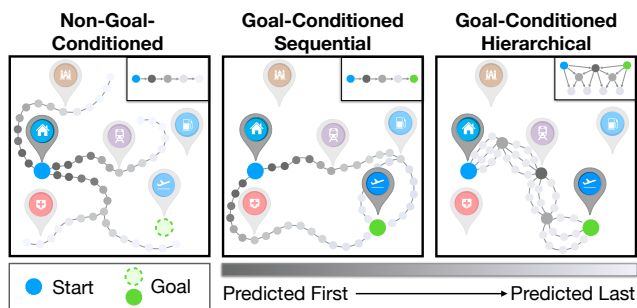

Figure 1: When planning towards faraway goals, we propose to condition the prediction of candidate trajectories on the goal, which significantly reduces the search space of possible trajectories (**left** vs. **middle**) and enables hierarchical planning approaches that break a long-horizon task into a series of short-horizon tasks by placing subgoals (**right**).

---

[*] Equal contribution. Ordering determined by a coin flip. Project page: `orybkin.github.io/video-gcp`

approaches do not scale to complex long-horizon tasks [11]. Imagine the task of planning a route from your home to the airport. The above approaches would attempt to model all possible routes starting at home and then search for those that ended up at the airport. For long-horizon problems, the number of possible trajectories grows very large, making extensive search infeasible.

In contrast, we propose a planning agent that only considers trajectories that start at home and end at the airport, i.e., makes predictions with the goal in mind. This approach both reduces prediction complexity as a simpler trajectory distribution needs to be modeled, and significantly reduces the search space for finding the best route, as depicted in Fig. 1 (center). Indeed, we can produce a feasible plan simply as a single forward pass of the generative model, and can further refine it to find the optimal plan through iterative optimization.

However, modeling this distribution becomes challenging for long time horizons even with goal-conditioned predictors. A naive method inspired by sequential predictive approaches would predict future trajectories at a fixed frequency, one step at a time — the equivalent of starting to plan the route to the airport by predicting the very first footsteps. This can lead to large accumulating errors. Moreover, the optimization problem of finding the best trajectory remains challenging. The sequential planning approaches are unable to focus on large important decisions as most samples are spent optimizing local variation in the trajectory. To alleviate both shortcomings, we propose to predict an a tree-structured way, starting with a coarse trajectory and recursively filling in finer and finer details. This is achieved by recursive application of a single module that is trained to answer: given two states, what is a state that occurs between them? This hierarchical prediction model is effective at long-term prediction and further enables us to design an efficient long-horizon planning approach by employing a coarse-to-fine trajectory optimization scheme.

Hierarchical prediction models naturally lend themselves to modeling the hierarchical structure present in many long-horizon tasks by breaking them into their constituent steps. However such procedural steps do not all occur on a regularly spaced schedule or last for equal lengths of time. Therefore, we further propose a version of our model based on a novel probabilistic formulation of dynamic time warping [56] that allows the model to select which frames to generate at each level in the tree, enabling flexible placement of intermediate predictions.

In summary, the contributions of this work are as follows. First, we propose a framework for goal-conditioned prediction and planning that is able to scale to visual observations by using a latent state model. Second, we extend this framework to hierarchical prediction and planning, which improves both efficiency and performance through the coarse-to-fine strategy and effective parallelization. We further extend this method to modeling the temporal variation in subtask structure. Evaluated on a complex visual navigation task, our method scales better than alternative approaches, allowing effective control on tasks longer than possible with prior visual planning methods.

## 2   Related Work

**Video interpolation.** We propose a latent variable goal-conditioned prediction model that is able to handle high-dimensional image observations. This resembles prior work on video-interpolation where given a start and a goal image, images are filled in between. So far such work has focused on short-term interpolation, often using models based on optical flow [40, 27, 46, 47]. Recent work has proposed neural network models that predict images directly, but this work still evaluates on short-horizon prediction [66]. The models introduced in our work by contrast scale to video sequences of up to 500 time steps modelling complex distributions that exhibit multi-modality.

**Visual planning and control.** Most existing visual planning methods [15, 50, 69, 11, 20] use model predictive control, computing plans forward in time by sampling state or action sequences. This quickly becomes computationally intractable for longer horizon problems, as the search complexity grows exponentially with the number of time steps [11]. Instead, we propose a method for goal-conditioned hierarchical planning, which is able to effectively scale to long horizons as it both reduces the search space and performs more efficient hierarchical optimization. Ichter and Pavone [23] also perform goal-conditioned planning by constraining the search space to trajectories that end at the goal, however, the method is only validated on low-dimensional states. In this paper, we leverage latent state-space goal-conditioned predictors that scale to visual inputs and further improve the planning by using a hierarchical divide-and-conquer scheme. Other types of goal-conditioned control include inverse models and goal-conditioned imitative models [49, 70, 62, 59]. However, these methods rely

on imitation learning and are limited to settings where high-quality demonstrations are available. In contrast, our goal-conditioned planning and control method is able to optimize the trajectory it executes, and does not require optimal training data.

**Hierarchical planning.** While hierarchical planning has been extensively explored in symbolic AI [55, 35, 29], these approaches are unable to cope with raw (e.g., image-based) observations, limiting their ability to solve diverse real-world tasks. Instead, we propose an approach that learns to perform hierarchical planning directly in terms of sensory observations, purely from data. Since our method does not require human-designed specification of tasks and environment, it is applicable in general settings where trajectory data can be collected. Recently, a number of different hierarchical planning approaches have been introduced [26, 51, 14, 44, 31, 45] that only work well with one or two layers of abstraction due to the architectural design or computational bottlenecks. Some of the few hierarchical planning approaches that have been shown to work with more than two layers of abstraction use tree-structured models [5, 28, 48]. However these models have not been shown to scale to high-dimensional spaces such as images. While also using a tree-structured model similar to our method, Chen et al. [5] make the assumption that the map in the physical workspace is known. To the best of our knowledge, our proposed hierarchical planning algorithm is the first to use a variable number of abstraction layers while scaling to high-dimensional inputs such as images.

# 3  Goal-Conditioned Prediction

In this section, we formalize the goal-condition prediction problem, and propose several models for goal-conditioned prediction, including both auto-regressive models and tree-structured models. To define the goal-conditioned prediction problem, consider a sequence of observations $[o_1, o_2, ...o_T]$ of length $T$. Standard forward prediction approaches (Fig 2, left) observe the first $k$ observations and synthesize the rest of the sequence. That is, they model $p(o_{k+1}, o_{k+2}, \ldots o_{T-1}|o_1, o_2, \ldots o_k)$. Instead, we would like our goal-conditioned predictors to produce intermediate observations given the first and last elements in the sequence (Fig 2, center and right). In other words, they must model $p(o_2, o_3, \ldots o_{T-1}|o_1, o_T)$. We propose several designs for goal-conditioned predictors that operate in learned compact state spaces for scalability and accuracy.

## 3.1  Goal-Conditioned Sequential Prediction

We first present a naive design for goal-conditioned prediction based on forward auto-regressive models. Autoregressive models operating directly on observations scale poorly in terms of computational efficiency and predictive performance [8, 4, 20]. We design a latent state-space model (GCP-sequential, shown in Fig 2, center) that predicts in a latent space represented by a random variable $s_t$ and then decodes the observations with a decoder $p(o_t|s_t)$. The latent state $s_t$ is used to allow handling partially observable settings. The likelihood of this model factorizes as follows:

$$p(o_2, o_3, \ldots o_{T-1}|o_1, o_T) = \int p(o_2|s_2)p(s_2|o_1, o_T) \prod_{t=3}^{T-1} p(o_t|s_t)p(s_t|s_{t-1}, o_1, o_T)ds_{2:T-1}. \quad (1)$$

We show in Sec 3.4 that this model is simple to implement, and can build directly on previously proposed auto-regressive sequence prediction models. However, its computational complexity scales with the sequence length, as every state must be produced in sequence. As we show empirically, this approach also struggles with modeling longer sequences due to compounding errors, and is prone to ignoring the goal information on these longer sequences as very long-term dependencies have to be modeled when predicting the second observation from the first observation and the goal.

## 3.2  Goal-Conditioned Prediction by Recursive Infilling

In order to scale goal-conditioned prediction to longer time horizons we now design a tree-structured GCP model that is both more efficient and more effective than the naive sequential predictor.

Suppose that we have an intermediate state prediction operator $p(s_t|\mathrm{pa}(t))$ that produces an intermediate latent state $s_t$ halfway in time between its two parent states $\mathrm{pa}(t)$. Then, consider the following alternative process for goal-conditioned prediction depicted in Fig 2 (right): at the beginning, the observed first and last observation are encoded into the latent state space as $s_1$ and $s_T$, and the prediction operator $p(s_t|\mathrm{pa}(t))$ generates $s_{T/2}$. The same operator may now be applied to two new

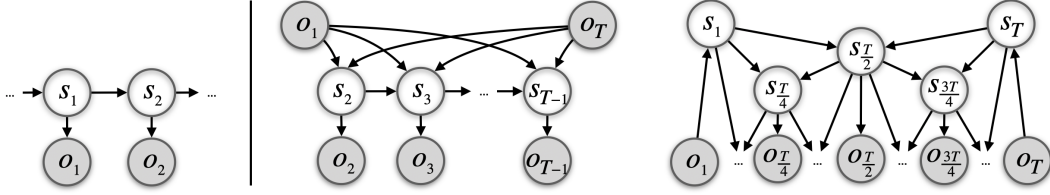

Figure 2: Graphical models for state-space sequence generation: forward prediction (left) and the proposed goal-conditioned predictors (GCPs). Shaded circles denote observations, white circles denote unobserved latent states. Center: a sequential goal-conditioned predictor with structure similar to forward prediction. Right: a hierarchical goal-conditioned predictor that recursively applies an infilling operator to generate the full sequence. All our models leverage stochastic latent states in order to handle complex high-dimensional observations.

sets of parents $(s_1, s_{T/2})$ and $(s_{T/2}, s_T)$. As this process continues recursively, the intermediate prediction operator fills in more and more temporal detail until the full sequence is synthesized.

We call this model GCP-tree, since it has a tree-like[1] shape where each predicted state is dependent on its left and right parents, starting with the start and the goal. GCP-tree factorizes the goal-conditioned sequence generation problem as:

$$p(o_2, o_3, \dots o_{T-1}|o_1, o_T) = \int p(s_1|o_1)p(s_T|o_T) \prod_{t=2}^{T-1} p(o_t|s_t)p(s_t|\mathrm{pa}(t))ds_{1:T}. \tag{2}$$

**Adaptive binding.** We have thus far described the intermediate prediction operator as always generating the state that occurs halfway in time between its two parents. While this is a simple and effective scheme, it may not correspond to the natural hierarchical structure in the sequence. For example, in the navigation example from the introduction, we might prefer the first split to correspond to visiting the bank, which partitions the prediction problem into two largely independent halves. We then design a version of GCP-tree that allows the intermediate state predictor to select which of the several states between the parents to predict, each time it is applied. In other words, the predicted state might *bind* to one of many observations in the sequence. In this more versatile model, we represent the time steps of the tree nodes with discrete latent variable $w$ that selects which nodes bind to which observations: $p(o_t|s_{1:N}, w_t) = p(o_t|s_{w_t})$. We can then express the prediction problem as:

$$p(o_{2:T-1}|o_1, o_T) = \int p(s_1|o_1)p(s_N|o_T) \prod_{n} p(s_n|\mathrm{pa}(n)) \prod_{t=2}^{T-1} p(o_t|s_{1:N}, w_t)p(w_t)ds_{1:N}dw_{2:T-1}.$$

Appendix F shows an efficient inference procedure for $w$ based on a novel probabilistic version of dynamic time warping [56].

### 3.3 Latent Variable Models for GCP

We have so far described the latent state $s_t$ as being a monolithic random variable. However, an appropriate design of $s_t$ is crucial for good performance: a purely deterministic $s_t$ might not be able to model the variation in the data, while a purely stochastic $s_t$ might lead to optimization challenges. Following prior work [8, 20], we therefore divide $s_t$ into $h_t$ and $z_t$, i.e. $s_t = (h_t, z_t)$, where $h_t$ is the deterministic memory state of a recurrent neural network, and $z_t$ is a stochastic per-time step latent variable. To optimize the resulting model, we leverage amortized variational inference [33, 52] with an approximate posterior $q(\tilde{z}|o_{1:T})$, where $\tilde{z} = z_{2:T-1}$. The deterministic state $h_t$ does not require inference since it can simply be computed from the observed data $o_1, o_T$. The training objective is the following evidence lower bound on the log-likelihood of the sequence:

$$\ln p(o_{2:T-1}|o_{1,T}) \geq \mathbb{E}_{q(\tilde{z})}\left[\ln p(o_{2:T-1}|o_{1,T}, \tilde{z})\right] - \mathrm{KL}\left(q(\tilde{z}) \,||\, p(\tilde{z}|o_{1,T})\right). \tag{3}$$

### 3.4 Architectures for Goal-Conditioned Prediction

We describe how GCP models can be instantiated with deep neural networks to predict sequences of high-dimensional observations $o_{1:T}$, such as videos. The prior $p(z_t|\mathrm{pa}(t))$ is a diagonal Gaussian whose parameters are predicted with a multi-layer perceptron (MLP). The deterministic state predictor $p(h_t|z_t, \mathrm{pa}(t))$ is implemented as an LSTM [22]. We condition the recurrent predictor on the start and goal observations encoded through a convolutional encoder $e_t = E(o_t)$. The decoding distribution $p(o_t|s_t)$ is predicted by a convolutional decoder with input features $\hat{e}_t$ and skip-connections from the encoder [64, 8]. In line with recent work [54], we found that learning a calibrated decoder is important for good performance, and we use the discrete logistics mixture as the decoding distribution [57]. The parameters of the diagonal Gaussian posterior distribution for each node, $q(z_t|o_t, \mathrm{pa}(t))$, are predicted given the corresponding observation and parent nodes with another MLP. For a more detailed description of the architectural parameters we refer to Appendix C.

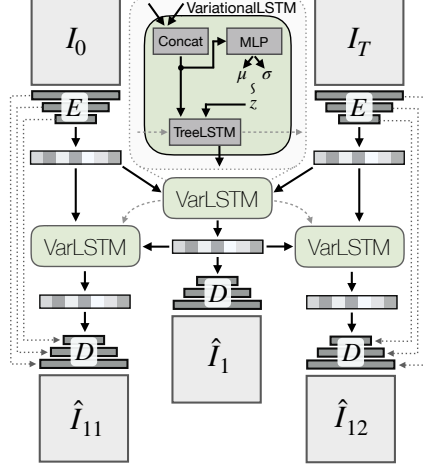

Figure 3: Architecture for two-layer hierarchical goal-conditioned predictor (GCP). Skip connections to first node's decoder omitted for clarity.

## 4 Planning & Control with Goal-Conditioned Prediction

In the previous section, we described an approach to goal-conditioned sequence prediction or GCP. The GCP model can be directly applied to control problems since, given a goal, it can produce realistic trajectories for reaching that goal. However, in many cases our objective is to reach the goal *in a specific way*. For instance, we might want to spend the least amount of time or energy required to reach the goal. In those cases, explicit planning is required to obtain a trajectory from the model that optimizes a user-provided cost function $\mathcal{C}(o_t, \ldots, o_{t'})$. In GCPs, planning is performed over the latent variables $z$ that determine *which* trajectory between start and goal is predicted: $\min_z \mathcal{C}(g(o_t, o_T, z))$, where $g$ is the GCP model. We propose to use the cross-entropy method (CEM, [53]) for optimization, which has proven effective in prior work on visual MPC [11, 42, 45, 51]. Once a trajectory is planned, we infer the actions necessary to execute it using a learned inverse model (see Appendix, Section E).

**Goal-conditioned hierarchical planning.** Instead of optimizing the full trajectory at once, the hierarchical structure of the GCP-tree model allows us to design a more efficient, hierarchical planning scheme in which the trajectories between start and goal are optimized in a coarse-to-fine manner. The procedure is detailed in Algorithm 1. We initialize the plan to consist of only start and goal observation. Then our approach recursively adds new subgoals to the plan, leading to a more and more de-

---

**Algorithm 1** Goal-Conditioned Hierarchical Planning

1: **Inputs:** Hierarchical goal-conditioned predictor $g$, current & goal observation $o_t, o_T$, cost function $\hat{\mathcal{C}}$
2: Initialize plan: $P = [o_t, o_T]$
3: **for** $d = 1...D$ **do**     ▷ iterate depth of hierarchy
4:     **for** $n = 0...|P| - 1$ **do**
5:         $\mathbf{z} \sim \mathcal{N}(0, I)$     ▷ sample M subgoal latents
6:         $\mathbf{o}_{\mathrm{sg}} = g(P[n], P[n+1], \mathbf{z})$     ▷ predict subgoals
7:         $o_{d,n} = \arg\min_{o \in \mathbf{o}_{\mathrm{sg}}} \hat{\mathcal{C}}(P[n], o) + \hat{\mathcal{C}}(o, P[n+1])$
8:         INSERT($P, o_{d,n}$)     ▷ insert best subgoal in plan
9: **return** $P$

---

tailed trajectory. Concretely, we proceed by optimizing the latent variables of the GCP-tree model $g(o_t, o_T, z)$ layer by layer: in every step we sample $M$ candidate latents per subgoal in the current layer and pick the corresponding subgoal that minimizes the total cost with respect to both its parents. The best subgoal gets inserted into the plan between its parents and the procedure recurses.

**Cost function.** Evaluating the true cost function $\mathcal{C}(o_t, \ldots, o_{t'})$ would require unrolling the full prediction tree. For more efficient, *hierarchical* planning, we instead want to evaluate the *expected* cost $\hat{\mathcal{C}}(o_t, o_{t'}) = \mathbb{E}_{(o_t,\ldots,o_{t'}) \sim \mathcal{D}} \mathcal{C}(o_t, \ldots, o_{t'})$ of a trajectory between two observations under the training data distribution $\mathcal{D}$. This allows us to estimate the cost of a trajectory passing through a given subgoal as the sum of the pairwise cost estimates to both its parents, without predicting all its children. We train a neural network estimator for the expected cost via supervised learning by randomly sampling two observations from a training trajectory and evaluating the true cost on the connecting trajectory segment $\mathcal{C}(o_t, \ldots, o_{t'})$ to obtain the target value.

Table 1: Long-term prediction performance of the goal-conditioned predictors compared to prior work on video interpolation. Additional evaluation on FVD / LPIPS [63, 72] in Appendix, Table 5.

| DATASET | PICK&PLACE | | HUMAN 3.6M | | 9 ROOMS NAV | | 25 ROOMS NAV | |
|---|---|---|---|---|---|---|---|---|
| METHOD | PSNR | SSIM | PSNR | SSIM | PSNR | SSIM | PSNR | SSIM |
| GCP-TREE | **34.34** | **0.965** | **28.34** | **0.928** | **13.83** | **0.288** | **12.88** | **0.279** |
| GCP-SEQUENTIAL | **34.45** | **0.965** | 27.57 | 0.924 | 12.91 | 0.213 | 11.61 | 0.209 |
| DVF [40] | 26.15 | 0.858 | 26.74 | 0.922 | 11.678 | 0.22 | 11.34 | 0.172 |
| CIGAN [36] | 21.16 | 0.613 | 16.89 | 0.453 | 11.96 | 0.222 | 9.91 | 0.150 |

## 5 Experimental Evaluation

The aim of our experiments is to study the following questions: (1) Are the proposed GCPs able to effectively predict goal-directed trajectories in the image space and scale to long time horizons? (2) Is the proposed goal-conditioned hierarchical planning method able to solve long-horizon visual control tasks? (3) Does the version of GCP with adaptive binding find high-level events in the trajectories?

### 5.1 Goal-Conditioned Video Prediction

Most commonly used video datasets in the literature depict relatively short motions, making them poorly suited for studying long-horizon prediction capability. We therefore evaluate on one standard dataset, and two synthetic datasets that we designed specifically for evaluating long-horizon prediction. The pick&place dataset contains videos of a simulated Sawyer robot arm placing objects into a bin. Training trajectories contain up to 80 frames

Table 2: Ablation of prediction performance on pick&place

| METHOD | PSNR | SSIM |
|---|---|---|
| TREE | 34.34 | 0.965 |
| TREE W/O SKIPS | 32.64 | 0.955 |
| TREE W/O LSTM | 31.44 | 0.947 |

at $64 \times 64$ px and are collected using a simple rule-based policy. The *Navigation* data consists of videos of an agent navigating a simulated environment with multiple rooms: we evaluate versions with 9-room and 25-room layouts, both of which use $32 \times 32$ px agent-centric top-down image observations, with up to 100 and 200 frame sequences, respectively. We collect example trajectories that reach goals in a randomized, suboptimal manner, providing a very diverse set of trajectories (details are in App. D). We further evaluate on the real-world Human 3.6M video dataset [25], predicting $64 \times 64$ px frames at full frequency of 50Hz up to 10 seconds in the future to show the scalability of our method. This is in contrast to prior work which evaluated on subsampled sequences shorter than 100 frames (see [9, 8, 67]). Architecture and hyperparameters are detailed in Appendix C.

In Tab. 1, we compare the GCP models to a state-of-the-art deep video interpolation method, DVF [40],[4] as well as a method for goal-conditioned generation of visual plans by interpolation in a learned latent space, CIGAN [36]. Following the standard procedure for evaluation of stochastic prediction models, we report top-of-100 peak signal-to-noise ratio (PSNR) and structural similarity met-ric (SSIM). We observe that the interpolation methods fail

Table 3: GCP runtime on $16 \times 16$ px H3.6M sequences in sec/training batch[3]

| SEQ LENGTH | 100 | 500 | 1000 |
|---|---|---|---|
| GCP-SEQ | 1.49 | 8.44 | 17.6 |
| GCP-TREE | **0.55** | **1.66** | **2.77** |
| SPEED-UP | ×2.7 | ×5.1 | ×6.4 |

to learn meaningful long-term dynamics, and instead blend between start and goal image or predict physically implausible changes in the scene. In contrast, GCP-sequential and GCP-tree, equipped with powerful latent variable models, learn to predict rich scene dynamics between distant start and goal frames (see qualitative results in Fig. 4 and for all methods on the project website.

On the longer Human 3.6M and 25-room datasets, the GCP-tree model significantly outperforms the GCP-sequential model. Qualitatively, we observe that the sequential model struggles to take into account the goal information on the longer sequences, as this requires modeling long-term dependencies, while the hierarchical model is able to naturally incorporate the goal information in the recursive infilling process. Additionally, the hierarchical structure of GCP-tree enables substantially

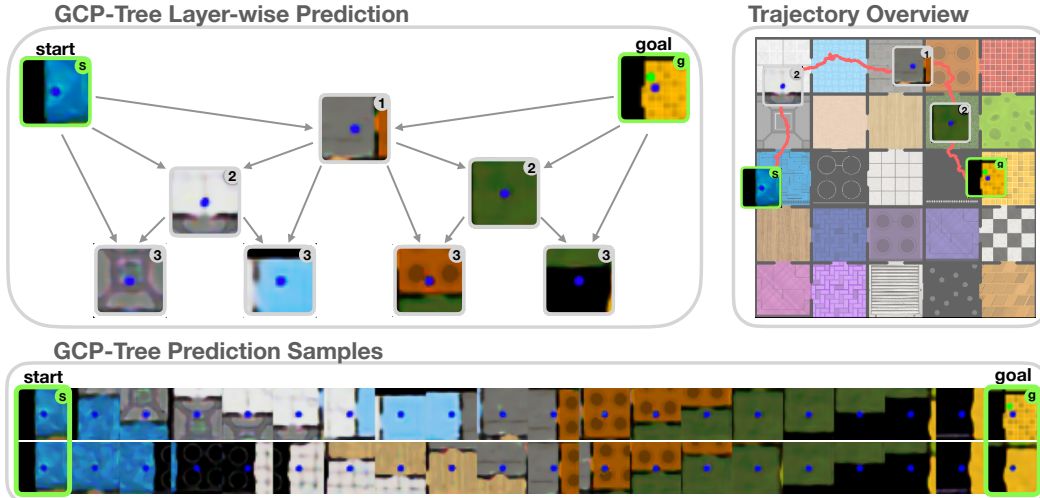

Figure 4: Samples from GCP-tree on the 25-room data. **Left**: hierarchical prediction process. At each layer, the infilling operator is applied between every two frames, producing a sequence with a finer and finer temporal resolution. Three layers out of eight are shown. **Right**: visualization of the trajectory on the map together with a plan execution (see Section 5.2). **Bottom**: two image sequences sampled given the same start and goal (subsampled to 20 frames for visualization). Our model leverages stochastic latent states that enable modeling multimodal trajectories. See additional video results on the supplementary website: `orybkin.github.io/video-gcp`.

faster runtimes (see Table 3). We present an ablation study for GCP-tree in Tab. 2, showing that both the skip connections and the recurrence in the predictive module contribute to good performance.

## 5.2 Visual Goal-Conditioned Planning and Control

Next, we evaluate our hierarchical goal-conditioned planning approach (see Section 4) on long-horizon visual control tasks. We test our method on a challenging image-based navigation task in the 9 and 25-rooms environments described in Section 5.1. We use the data from Section 5.1 to train the predictive model. We

Table 4: Visual control performance on navigation tasks

| METHOD | 9-ROOM NAV | | 25-ROOM NAV | |
|---|---|---|---|---|
| | SUCCESS | COST | SUCCESS | COST |
| GC BC [43] | 45% | 139.75 | 7% | 402.48 |
| VF [11] | 84% | 128.00 | 26% | 362.82 |
| OURS | **93**% | **34.34** | **82**% | **158.06** |
| GCP-FLAT | **94**% | 36.00 | 79% | 181.02 |
| GCP-SEQUENTIAL | 91% | 50.02 | 14% | 391.99 |

note that our method does not require optimal demonstrations, but only data that is sufficiently diverse. Such dataset might be collected e.g. via crowd-sourced teleoperation [73], or with a suitable exploration policy [12, 58]. For evaluation with purely random data, see supp. section G. Given the current image observation the agent is tasked to reach the goal, defined by a goal image, on the shortest possible path. We average performance over 100 task instances for evaluation. These tasks involve crossing up to three and up to 10 rooms respectively, requiring planning over horizons of several hundred time steps, much longer than in previous visual planning methods [10, 11].

We compare hierarchical planning with GCP to visual foresight (VF, Ebert et al. [11]), which optimizes rollouts from an action-conditioned forward prediction model via CEM [53]. We adopt the improvements to the sampling and CEM procedure introduced in Nagabandi et al. [42]. We also compare to goal-conditioned behavioral cloning (GC BC, [43]) as a "planning-free" approach for learning goal-reaching from example goal-reaching behavior.

In Table 4, we report the average success rate of reaching the goal room, as well as the average cost, which corresponds to the trajectory length.[5] VF performs well on the easy task set, which requires

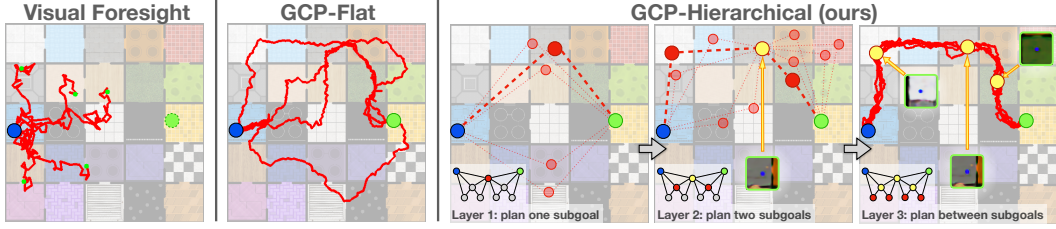

Figure 5: Comparison between planning methods. Trajectories (red) sampled while planning from start (blue) to goal (green). All methods predict image trajectories, which are shown as 2d states for visualization. **Left**: visual MPC [11] with forward predictor, **middle**: non-hierarchical planning with goal-conditioned predictor (GCP), **right**: hierarchical planning with GCP (ours) recursively optimizes subgoals (yellow/red) in a coarse-to-fine manner and finally plans short trajectories between the subgoals. Goal-conditioning ensures that trajectories reach the long-horizon goal, while hierarchical planning decomposes the task into shorter segments which are easier to optimize.

planning horizons similar to prior work on VF, but struggles on the longer tasks as the search space becomes large. The BC method is not able to model the complexity of the training data and fails to solve these environments. In contrast, our approach performs well even on the long-horizon task set.

We compare different planning approaches in Fig. 5. We find that samples from the forward prediction model in VF have low probability of reaching long-horizon goals. Using GCPs with a non-hierarchical planning scheme similar to [11, 42] (GCP-Flat) requires optimization over a large set of possible trajectories between start and goal and can struggle to find a plan with low cost. In contrast, our hierarchical planning approach finds plans with low cost by breaking the long-horizon task into shorter subtasks through multiple recursions of subgoal planning. Using GCP-sequential instead of GCP-tree for sampling performs well on short tasks, but struggles to scale to longer tasks (see Table 4), highlighting the importance of the hierarchical prediction model.

## 5.3 Temporal Abstraction Discovery

We qualitatively evaluate the ability of GCP-tree *with adaptive binding* (see Section 3.2) to learn the temporal structure in the robotic pick&place dataset. We increase the reconstruction loss of the nodes in the first two layers of the tree 50 times, forcing these nodes to bind to the frames for which the prediction is the most confident, the bottlenecks (see experimental details in Appendix F).

In Fig. 6, we see that this structural prior causes the model to bind the top nodes to frames that represent semantic bottlenecks, e.g. when the robot is about to drop the object in the bin. We found that all three top layer nodes specialize on binding to distinctive bottlenecks, leading to diverse predicted tree structures. We did not observe that adaptive binding improves the quality of predictions on our datasets, though the ability to discover meaningful bottlenecks may itself be useful [3, 34, 17].

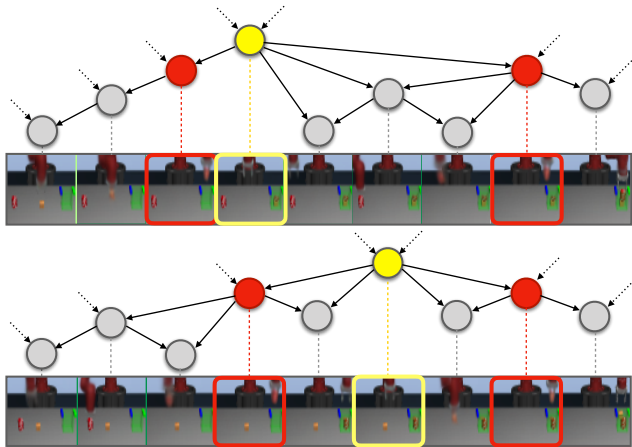

Figure 6: Temporal abstraction discovery on pick&place. Discovered tree structure with adaptive binding: nodes from the first two layers (yellow/red) bind to semantically consistent bottlenecks across sequences, e.g. in which the robot is about to drop the object into the bin.

# 6 Discussion

We present two models for goal-conditioned prediction: a standard sequential architecture and a hierarchical tree-structured variant, where the latter either splits the sequence into equal parts at each level of the tree, or into variable-length chunks via an adaptive binding mechanism. We further propose an efficient hierarchical planning approach based on the tree-structure model. All variants of our method outperform prior video interpolation methods, and the hierarchical variants substantially outperform the sequential model and prior visual MPC approaches on a long-horizon image-based navigation task. Additionally, the adaptive binding model can discover bottleneck subgoals.

## Broader Impact

We proposed a method for visual prediction and planning that is able to solve long-horizon tasks autonomously. This method may have a broader impact on capabilities of robots performing tasks such as autonomous navigation or object manipulation, and may be applicable in settings such as navigation of zones dangerous for humans, search and rescue, as well as warehouse robotics applications. While the method, and in general all planning and reinforcement learning methods, may be applied to a variety of settings, including those with questionable ethical motivation, we are optimistic of the general positive impact of future autonomous robotic systems, especially in the areas described above.

Another ethical consideration is that, since the model is able to produce long videos targeted to a particular goal, it might be used to produce fake videos of people performing a certain action, and provides a degree of control about that action through the specification of the goal image. This might enable forging fake videos targeted at specific persons. However, recent research has shown that most current methods for generating fake videos are easily detectable, both by people and automatic detection methods [18, 1, 38].

## Acknowledgements

We thank Suraj Nair, Thanard Kurutach, and Aviral Kumar for fruitful discussions. We would like to thank Ben Eysenbach, Ayush Jain and two anonymous internal reviewers for feedback on an earlier version of the paper, Shenghao Zhou for discussion and help with preliminary evaluation of the method, and Kristian Hartikainen for discussion and software development tips. Support was provided by the ARL DCIST CRA W911NF-17-2-0181 grant, and by Honda Research Institute. KP and OR were visitors at UC Berkeley while conducting this research.

## Footnotes

[1]The generation process closely mimics a graph-theoretic tree, but every node has two parents instead of one.

[3]We use $16 \times 16$ px to fit the 1000-frame sequences on a single NVIDIA V100 GPU, we expect the results to translate to larger resolutions on GPUs with larger memory.

[4]While DVF has an official trained model, we re-train DVF on each dataset for better performance.

[5]Since reporting length for failed cases would skew the results towards methods that produce short, unsuccessful trajectories, we report a constant large length for failed trajectories.

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
