[Supplementary Material]



Figure 7: Predictions on Human 3.6M. We see that the GCP models are able to faithfully capture the human trajectory. The optical flow-based method (DVF) captures the background but fails to generate complex motion needed for long-term goal-conditioned prediction. Causal InfoGan also struggles to capture the structure of these long sequences and produce implausible interpolations. Full qualitative results are on the supplementary website: `sites.google.com/view/gcp-hier/home`.

Table 5: Prediction performance on perceptual metrics.

| DATASET | PICK&PLACE | | HUMAN 3.6M | | 9-ROOM MAZE | | 25-ROOM MAZE | |
|---|---|---|---|---|---|---|---|---|
| METHOD | FVD | LPIPS | FVD | LPIPS | FVD | LPIPS | FVD | LPIPS |
| GCP-TREE | 430.3 | **0.02** | **1314.3** | 0.05 | **655.50** | **0.174** | **413.31** | **0.168** |
| GCP-SEQUENTIAL | **328.9** | **0.02** | 1541.8 | **0.06** | 860.04 | 0.214 | 638.95 | 0.238 |
| DVF [40] | 2879.9 | 0.06 | 1704.6 | **0.05** | 1320.34 | 0.231 | 1476.44 | 0.215 |
| CIGAN [36] | 3252.6 | 0.12 | 2528.5 | 0.17 | 1440.6 | 0.190 | 677.40 | 0.219 |

# A  Additional results

We include additional qualitative and quantitative results here as well as at the supplementary website: `sites.google.com/view/video-gcp`.

# B  Evidence lower bound (ELBO) derivation

We wish to optimize the likelihood of the sequence conditioned on the start and the goal frame $p(o_{2:T-1}|o_{1,T})$. However, due to the use of latent variable models, this likelihood is intractable, and we resort to variational inference to optimize it. Specifically, we introduce an approximate posterior network $q(z_{2:T-1}|o_{1:T})$, where that approximates the true posterior [33, 52]. The ELBO can be derived from the objective that consists of likelihood and a term that enforces that the approximate posterior matches the true posterior:

$$\ln p(o_{2:T-1}|o_{1,T}) \geq \ln p(o_{2:T-1}|o_{1,T}) - \mathrm{KL}(q(z_{2:T-1}|o_{1:T})||p(z_{2:T-1}|o_{1:T}))$$
$$= \mathbb{E}_{q(z_{2:T-1}|o_{1:T})}\left[\ln p(o_{2:T-1}|o_{1,T}, z_{2:T-1})\right] - \mathrm{KL}\left(q(z_{2:T-1}|o_{1:T})\,||\,p(z_{2:T-1}|o_{1,T})\right), \quad (4)$$

where the last equality is simply a rearrangement of terms.

Figure 8: Prior samples from GCP-tree on the Human 3.6M dataset. Each row is a different prior sample conditioned on the same information.

Further, in order to efficiently parametrize these distributions, we factorize the distributions as follows according to the graphical model in Fig 2 (right) and Eq. 2:

$$p(o_{2:T-1}|o_{1,T}, z_{2:T-1}) = \prod_{t=2}^{T-1} p(o_t|o_{1,T}, z_t), \tag{5}$$

$$p(z_{2:T-1}|o_{1,T}) = \prod_{t=2}^{T-1}, \tag{6}$$

$$q(z_{2:T-1}|o_{1:T}) = \prod_{t=2}^{T-1} q(z_t|o_t, \text{pa}(t)). \tag{7}$$

We therefore require the following distributions to define our model: $p(o_t|o_{1,T}, z_t)$, $p(z_t|\text{pa}(t))$, $q(z_t|o_t, \text{pa}(t))$. The parameterization of these distributions is defined in Section 3.4. The parent operator $\text{pa}(t)$ returns the parent nodes of $s_t$ according to the graphical model in Fig 2 (right). Using these factorized distributions, we can write out the ELBO in more detail as:

$$\ln p(o_{2:T-1}|o_{1,T}) \geq \mathbb{E}_{q(z_{2:T-1}|o_{1:T})} \sum_{t=2}^{T-1} \left[\ln p(o_t|o_{1,T}, z_t) - \text{KL}\left(q(z_t|o_t, \text{pa}(t)) \,||\, p(z_t|\text{pa}(t))\right)\right]. \tag{8}$$

## C Architecture

We use a convolutional encoder and decoder similar to the standard DCGAN discriminator and generator architecture respectively. The latent variables $z_n$ as well as $e_n$ are 256-dimensional. All hidden layers in the Multi-Layer Perceptron have 256 neurons. We add skip-connections from the encoder activations from the first image to the decoder for all images. For the inference network we found it beneficial to use a 2-layer 1D temporal convolutional network that adds temporal context into the latent vectors $e_t$. For the recursive predictor that predicts $e_n$, we use group normalization [68]. We found that batch normalization [24] does not work as well as group normalization for the recursive predictor and conjecture that this is due to the activation distributions being non-i.i.d. for different levels of the tree. We use batch normalization in the convolutional encoder and decoder, and use local per-image batch statistics at test time. Further, for the simple RNN (without the LSTM architecture)

Figure 9: Comparison of visual planning & control approaches. Execution traces of Visual Foresight (**left**), GCP-tree with non-hierarchical planning (**middle**) and GCP-tree with hierarchical planning (**right**) on two 25-room navigation tasks. Visualized are start and goal observation for all approaches as well as predicted subgoals for hierarchical planning. Both GCP-based approaches can reach faraway goals reliably, but GCP with hierarchical planning finds shorter trajectories to the goal.

ablation of our tree model, we activate $e_n$ with hyperbolic tangent (tanh). We observed that without this, the magnitude of activations can explode in the lower levels of the tree and conjecture that this is due to recursive application of the same network. We found that using TreeLSTM [60] as the backbone of the hierarchical predictor significantly improved performance over vanilla recurrent architectures.

To increase the visual fidelity of the generated results when predicting images, we use a foreground-background generation procedure similar to [65]. The decoding distribution $p(o_t|s_t)$ is a mixture of discretized logistics [57], which we found to work better than alternative distributions. We use the mean of the decoding distribution as the prediction.

For the adaptive binding model, the frame $o_t$ corresponding to the node $s_n$ is not known before the $s_n$ is produced. We therefore conditioned the inference distribution on the entire evidence sequence $o_{1:T}$ via the attention mechanism over the embeddings [2, 41]: $q(z_t) = \text{Att}(\text{enc}(o_{1:T}), \text{pa}(t))$. We reuse the same observation embeddings $e_t$ for the attention mechanism values.

The different paths between the same start and goal may have very different lengths (see e.g. Fig. 9), so it is necessary for GCP models to predict sequences of different lengths. We do so by training a termination classifier that predicts how long the sequence is. For GCP-Sequential, the termination classifier simply outputs the number of frames in the sequence, and the sequence is produced by recurrently unrolling that many frames. For the GCP-Tree model, to account for varied shapes of the tree, we instead predict a binary termination value at each node. To sample a trajectory, we recursively expand the tree, but stop the expansion where a particular node was classified as terminal (determined by a threshold on the classifier output). This procedure enables us to model even datasets with sequences of variable lengths.

**Hyperparameters.** The convolutional encoder and decoder both have five layers. We use the Rectified Adam optimizer [39, 32] with $\beta_1 = 0.9$ and $\beta_2 = 0.999$, batch size of 16 for GCP-sequential and 4 for GCP-tree, and a learning rate of $2e-4$. On each dataset, we trained each network for the same number of epochs on a single high-end NVIDIA GPU. Training took a day for all datasets except the 25-room dataset, where we train the models for 3 days.

## D  Data processing and generation

For training GCPs we use a dataset of example agent goal-reaching behavior. Below we describe how we collect those examples on the pick&place and navigation tasks and the details of the Human3.6M dataset. The data can be found on the following links:

- 9-room: `https://www.seas.upenn.edu/~oleh/datasets/gcp/nav_9rooms.zip`

- 25-room: `https://www.seas.upenn.edu/~oleh/datasets/gcp/nav_25rooms.zip`

- pick&place: `https://www.seas.upenn.edu/~oleh/datasets/gcp/sawyer.zip`

- Pre-processed H3.6: `https://www.seas.upenn.edu/~oleh/datasets/gcp/h36m.zip`

Figure 10: Example trajectory distributions between fixed start (red) and goal (green) rooms on the 25-room navigation task. The example goal-reaching behavior is highly suboptimal, with both strong multimodality in the space of possible solutions as well as low-level noise in each individual trajectory.

**pick&place.** We generate the pick&place dataset using the RoboSuite framework [13] that is based on the Mujoco physics simulator [61]. We generate example goal-reaching trajectories by placing two objects at random locations on the table and using a rule-based policy to move them into the box that is located at a fixed position on the right of the workspace. We sample the object type randomly from a set of two possible object types, bread and can, with replacement.

**Human 3.6M.** For the Human 3.6 dataset, we downsample the original videos to 64 by 64 resolution. We obtain videos of length of roughly 800 to 1600 frames, which we randomly crop in time to 500-frame sequences. We split the Human 3.6 into training, validation and test set by correspondingly 95%, 5% and 5% of the data.

**Navigation.** For the navigation task the agent is asked to plan and execute a path between a given 2D start and goal position. The environment is simulated using the Gym-Miniworld framework [6]. We collect goal-reaching examples by randomly sampling start and goal positions in the 2D maze and plan trajectories using the Probabilistic Roadmap (PRM, Kavraki et al. [30]) planner. The navigation problem is designed such that multiple possible room sequences can be traversed to reach from start to goal for any start and goal combination. During planning we sample one possible room sequence at random, but constrain the selection to only such sequences that do not visit any room more than once, i.e. that do not have loops. This together with the random sampling of waypoints of the PRM algorithm leads to collected examples of goal reaching behavior with substantial suboptimality. We show an example trajectory distribution from the data in Fig. 10. While GCPs support training on sequences of variable length we need to set an upper bound on the length of trajectories to bound the required depth of the hierarchical predictive model and allow for efficient batch computation (e.g. at most 200 frames for the 25-room environment). If plans from the PRM planner exceed this threshold we subsample them to the maximum lenght using spline interpolation before executing them in the environment. The training data consists of 10,000 and 23,700 sequences for the 9-room and the 25-room task respectively, which we split at a ration of 99%, 1%, 1% into training, validation and test.

Table 6: Hyperparameters for hierarchical planning with GCPs on 9-room and 25-room navigation tasks.

| Hierarchical Planning Parameters | |
|---|---|
| Hierarchical planning layers ($D$) | 2 |
| Samples per subgoal ($M$) | 10 |
| Final Segment Optimization | |
| Sequence samples per Segment | 5 |
| General Parameters | |
| Max. episode steps | 200 / 400 |
| Cost function | $\sum_{t=0}^{T-1}(x_{t+1} - x_t)^2$ |

# E    Planning Experimental Setup

For planning with GCPs we use the model architectures described in Section C trained on the navigation data described in Section D. The hyperparameters for the hierarchical planning experiments are listed in Table 6. We keep the hyperparameters constant across both 9-room and 25-room tasks except for the maximum episode length which we increase to 400 steps for the 25-room task. Note that the cost function is only used at training time to train the cost estimator described in Section 4, which we use to estimate all costs during planning.

To infer the actions necessary to execute a given plan, we train a separate inverse model $a_t = f_{\text{inv}}(o_t, o_{t+1})$ that infers the action $a_t$ which leads from observation $o_t$ to $o_{t+1}$. We train the inverse model with action labels from the training dataset and, in practice, input predicted feature vectors $\hat{e}_t$ instead of the decoded observations to not be affected by potential inaccuracies in the decoding process. We use a simple 3-layer MLP with 128 hidden units in each layer to instantiate $f_{\text{inv}}$. At every time step the current observation along with the next observation from the plan is passed to the inverse model and the predicted action is executed. We found it crucial to perform such closed-loop control to avoid accumulating errors that posed a central problem when inferring the actions for the whole plan once and then executing them open-loop.

We separately tuned the hyperparameters for the visual foresight baseline and found that substantially more samples are required to achieve good performance, even on the shorter 9-room tasks. Specifically, we perform three iterations of CEM with a batch size of 500 samples each. For sampling and refitting of action distributions we follow the procedure described in [42]. We use a planning horizon of 50 steps and replan after the current plan is executed. We cannot use the cost function from Table 6 for this baseline as it leads to degenerate solutions: in constrast to GCPs, VF searches over the space of *all* trajectories, not only those that reach the goal. Therefore, the VF planner could minimize the trajectory length cost used for the GCP models by predicting trajectories in which the agent does not move. We instead use a cost function that measures whether the predicted trajectory reached the goal by computing the L2 distance between the final predicted observation of the trajectory and the goal observation.

We run all experiments on a single NVIDIA V100 GPU and find that we need approximately 30mins / 1h to evaluate all 100 task instances on the 9-room and 25-room tasks respectively when using the hierarchical GCP planning. The VF evaluation requires many more model rollouts and therefore increases the runtime by a factor of approximately five, even though we increase the model rollout batch size by a factor of 20 for VF to parallelize trajectory sampling as much as possible.

# F    Adaptive Binding with Dynamic Programming

## F.1    An efficient inference procedure

To optimize the model with adaptive binding, we perform variational inference on both $w$ and $z$:

$$\log p(x) \geq \mathbb{E}_{q(z,w)}[p(x|w,z)] - D_{KL}(q(z|x)||p(z)) - D_{KL}(q(w|x,z)||p(w)). \qquad (9)$$

To infer $q(w|x, z)$, we want to produce a distribution over possible alignments between the tree and the evidence sequence. Moreover, certain alignments, such as the ones that violate the ordering of the sequence are forbidden. We define such distribution over aligment matrices $A$ via Dynamic Time Warping. We define the energy of an alignment matrix as the cost, and the following distribution over alignment matrices:

$$p(A|x, z) = \frac{1}{Z} e^{-A*c(x,z)},$$

where the partition function $Z = \mathbb{E}_A[e^{-A*c(x,z)}]$, and $c$ is the MSE error between the ground truth frame $x_t$ and the decoded frame associated with $z_n$. We are interested in computing marginal edge distributions $w = \mathbb{E}_A[A]$. Given these, we can compute the reconstruction error efficiently. We next show how to efficiently compute the marginal edge distributions.

Given two sequences $x_{0:T}, z_{0:N}$, denote the partition function of aligning two subsequences $x_{0:i}, z_{0:j}$ as $f_{i,j} = \sum_{A \in \mathcal{A}_{0:i,0:j}} e^{-A*c(x_{0:i},z_{0:j})}$. [7] shows that these can be computed efficiently as:

$$f_{i,j} = c(x_i, z_j) * (f_{i-1,j-1} + f_{i-1,j}).$$

We note that we do not include the third term $f_{i,j-1}$), as we do not want a single predicted frame to match multiple ground truth frames. Furthermore, denote the partition function of aligning $x_{i:T}, z_{j:N}$ as $b_{i,j} = \sum_{A \in \mathcal{A}_{i:T,j:N}} e^{-A*c(x_{i:T},z_{j:N})}$. Analogously, we can compute it as:

$$b_{i,j} = c(x_i, z_j) * (b_{i+1,j+1} + b_{i+1,j}).$$

**Proposition 1** *The total unnormalized density of all alignment matrices that include the edge $(i, j)$ can be computed as $e_{i,j} = f_{i,j} * b_{i,j}/c(x_i, z_j) = c(x_i, z_j) * (f_{i-1,j-1} + f_{i-1,j}) * (b_{i+1,j+1} + b_{i+1,j})$. Moreover, the probability of the edge $(i, j)$ can be computed as $w_{i,j} = e_{i,j}/Z$.*

Proposition 1 enables us to compute the expected reconstruction loss in quadratic time:

$$p(x|z) = w * c(x, z).$$

### F.2 Bottleneck Discovery Experimental Setup

In order to use the adaptive binding model to discover bottleneck frames that are easier to predict, we increase the reconstruction loss on those nodes as described in the main text. Specifically, we use Gaussian decoding distribution for this experiment, and set the variance of the decoding distribution for several top layers in the hierarchy to a fraction of the value for lower layers. This encourages the model to bind the frames that are easier to predict higher in the hierarchy as the low variance severely penalizes poor predictions. We found this simple variance re-weighting scheme effective at discovering bottleneck frames on several environments.

To generate the visualization of the discovered tree structure in Fig. 6 we evenly subsample the original 80-frame sequences and display those nodes that bound closest to the subsampled frames such that the resulting graph structure still forms a valid 2-connected tree. The variations in tree structure arise because the semantic bottlenecks which the nodes specialize on binding to appear at different time steps in the sequences due to variations in speed and initial position of the robot arm as well as initial placement of the objects.

## G Training from Random Data

In the room navigation experiments we train our model with noisy trajectories that reach diverse goals with considerable suboptimality (see Fig. 10). To test whether our method can work with even more suboptimal training data, we conduct preliminary experiments with completely random exploration data, and observe that our method still successfully solves navigation tasks in the 9-room environment (see Fig. 11). This suggests that the proposed method is scalable even to situations where no good planners exist that can be used for data collection.

Figure 11: **Left**: random exploration data.
**Right**: execution of our method trained on
random data.

Table 7: Average Trajectory Length. Planning with GCP finds shorter paths than the training
distribution.

| | ORIGINAL DATA | RANDOM DATA |
|---|---|---|
| TRAINING DATA | 31.4 | 62.6 |
| GCP-TREE (OURS) | **20.7** | **42.6** |

trajectory length of training data and our method on both, the dataset used for the experiments in section 5.1 and the random action data. We find that planning with our method leads to substantially shorter trajectories, further showing the ability of our approach to improve upon low-quality training data.

## G.1 Runtime Complexity

**Computational efficiency.**    While the sequential forward predictor performs $\mathcal{O}(T)$ sequential operations to produce a sequence of length T, the hierarchical prediction can be more efficient due to parallelization. As the depth of the tree is $\lceil \log T \rceil$, it only requires $\mathcal{O}(\log T)$ sequential operations to produce a sequence, assuming all operations that can be conducted in parallel are parallelized perfectly. We therefore batch the branches of the tree and process them in parallel at every level to utilize the benefit of efficient computation on modern GPUs. We note that the benefits of the GCP-tree runtime lie in parallelization, and thus diminish with large batch sizes, where the parallel processing capacity of the GPU is already fully utilized. We notice that, when predicting video sequences of 500 frames, GCP-sequential can use up to 4 times bigger batches than GCP-Tree without significant increase in runtime cost. This benefit is applicable both during training and inference.

When training tree-structured networks we exploit the provided parallelism in the structure of the model and batch recursions in the tree that are independent when conditioned on their parents.