[Reviews · NeurIPS 2020]

Review 1

Summary and Contributions: This paper studies two issues in sequential decision making. Most current approaches start planning from a start state, taking actions at the lowest temporal resolution. This paper studies 1) how to incorporate goal information in a plan, and 2) how to plan at a hierarchical level, from coarse to fine grained temporal resolution. The authors introduce goal-conditioned predictors, a class of models that inpaint a predicted state that lies between two given states. They add a latent state that separate deterministic and stochastic aspects of the prediction, and also add a dynamic time warping approach to adaptively select where to best predict in between two given states. They show improved planning performance on 25-room planning task over competing planning approaches from visual data. Update after rebuttal: - I really value the new results, where you show that the method can find solutions that improve over the demonstrations (and the fact that you included the actual results in the rebuttal, not just mentioned that you have new results). Do indeed include them in the final version. - The one thing you did not really answer is the issue of the z variable modelling both the effect of actions and environment stochasticity, in case of a stochastic environment. Optimizing over z will then also optimize over the environment stochasticity, which is of course not in your control. I believe you now focus on deterministic environments, but I would discuss this issue somewhere. - I stick with my strong accept vote.

Strengths: - The paper is well-written and clear. - The methodology is novel, interesting, and very relevant for the field. - The experimental work section asks the right questions for evaluation (does the prediction itself work, does planning over these predictions work, and does the adaptive binding work)

Weaknesses: - My main worry is that the training trajectories are by no means optimal. This will not matter much on short trajectories, but on long trajectories large detours can be more problematic. I understand that during planning you optimise over z to partially overcome this issue, but in larger problems I would worry that the optimisation over z becomes infeasible. I think you should at least identify this issue in the paper, and comment on it. - The z variables capture a combination of variation in actions and environment stochasticity. One can indeed be influenced, the other not. I think your current test environments are deterministic. Would you method run into trouble in stochastic environments? It would optimise over z variables as if it can influence stochastic aspects of the environment. - Table 1/2: People from the reinforcement learning community (like myself) are not familiar with the PSNR and SSIM metrics. Therefore, these tables are hard to interpret. Could you add some intuition about what a difference in SSIM of 0.928 versus 0.922 makes, for example? Minor - Table 4: What do “hier planning” and “hier prediction” mean? Do you mean “GPC tree” and “GPC sequential”? - Figure 3: This figure is really cropped, at first I did not really get that the VarLSTM part was an inlay. L85: For related work, I would be interested in how they enforce this. L109: goal-conditioned L54: an a

Correctness: Yes the claims and methods seem correct.

Clarity: Yes.

Relation to Prior Work: Yes the paper discusses related work in Sec. 2. Most previous work in hierarchical RL learns hierarchies from the start. This methods does require a known goal state, but then recursively breaks down the plan, which is a completely different approach (which indeed has much parallel to human planning as well).

Reproducibility: Yes

Additional Feedback: Conclusion: This is a well-written paper about an important challenge in sequential decision-making. The idea the authors present is interesting: given a start and goal-state, we can recursively plan a trajectory in between by repeatedly splitting it (approximately) in half. This is different from the standard approaches to hierarchical RL, and as such of interest to the community. The authors have extensively investigated this problem, for example also adding dynamic time warping to adaptively select the points to split the trajectory. The results outperform the state-of-the-art, and as a side effect their method can also detect bottlenecks. I think this paper has many interesting insights for the community, and would recommend it for acceptance.


Review 2

Summary and Contributions: This paper studies the goal-conditioned long-horizon visual prediction and planning tasks. (1) They formulated the goal-conditioned prediction problem and built a goal-conditioned predictor (GCP). (2) GCP can predict the future trajectories in a divide-and-conquer style which form a natural hierarchy. (3) In experiments, they show that such a model can help them solve long-horizon visual planning tasks. --- Update --- I read the author's rebuttal, and I appreciate the explanation. I will maintain my score and vote for acceptance.

Strengths: Combining visual prediction into reinforcement learning and using that to solve the long-horizon problem is a promising direction. Building the hierarchical model helps us to solve the long-horizon prediction tasks in the model-based RL. Applying TreeLSTM into the hierarchical model and using dynamic programming for adaptive binding are novel to the RL and control communities and will inspire future works. Especially, using Adaptive time binding to find bottleneck is a very interesting idea. This paper also includes several details to learn a better stochastic model. The whole pipeline is technically sound, and the experiments part convincingly shows the potential of their approaches.

Weaknesses: They only evaluated the planning in the 2D navigation domain. Does this mean that the proposed approach is not suitable for other visual planning tasks like pick&place? 2D navigation favors graph-based/map-based motion planning methods like [1] and may not be the best platform for studying visual planning tasks. I would appreciate it if the author can show the results on other challenging environments like Atari/Doom/Fetch Pick &Place. Besides, applying divide and conquer to build the subgoal tree in goal conditioned RL is not very novel considering [26], as mentioned in the paper. [1] Search on the Replay Buffer: Bridging Planning and Reinforcement Learning

Correctness: Yes

Clarity: Yes

Relation to Prior Work: Yes

Reproducibility: Yes

Additional Feedback: The proposed approach is related to trajectory optimization and motion planning field in optimal control. Sample-based motion planner sample random states and connects them as a graph. If we adaptively sample the states, we can also build a hierarchical graph and thus speed up the process. Can the author put some words on this? Comparing with the sequential model, the goal-conditioned predictor may suffer from the generalization issue. The sequential model only requires the local information and thus can sometimes generalize well (for example, in some physical prediction tasks), but suffers from the long-horizon prediction as it lacks the global constraints. At the same time, the goal-conditioned predictor needs to memorize the long-horizon trajectories, which brings challenge to the neural networks. I wonder if it is possible to unify the learned inverse model, the hierarchy goal-conditioned predictors and the single step forward model into the same framework for solving more challenging tasks. The adaptive binding does not improve the performance, but such an additional module should change something. How does the adaptive binding affect the performance? Will it make the performance worse?


Review 3

Summary and Contributions: This paper introduces a visual goal-conditioned planner that, instead of predicting the next frame, proposes subgoals in between the start state and the goal. It then recursively proposes subgoals between subgoals until a full path is proposed. It does this in latent space. The planner learns from examples of successful trajectories. ### After Author Rebuttal ### I have read the authors' rebuttal and I appreciate them addressing my concern about learning from suboptimal examples. This has given me a more positive outlook on the paper and I am changing my review from a 4 to a 6.

Strengths: The algorithm introduced in this paper addresses two shortcomings from visual prediction: 1) Errors compounding over time. Visual prediction usually does not work well after many timesteps because errors accumulate over time. However, this algorithm predicts an intermediate step between the start state and the goal, ensuring that it can still predict a high-quality image many steps into the future. 2) Having to predict irrelevant details. If a domain has irrelevant data (for example, television static), it can be hard to learn visual prediction because the predictor may be distracted by high errors on irrelevant data. Because the algorithm plans in a learned latent space, it can learn to model only relevant details in that latent space while ignoring irrelevant details.

Weaknesses: This algorithm relies on examples of trajectories that successfully solve the problem in order to learn. This would not be such a major weakness if the authors were more specific about the use-case of their algorithm. Given this requirement, this algorithm would not be applicable to cases where successful trajectories cannot be obtained. I think the authors should make this clear and identify what are plausible and currently implausible applications. The algorithm builds a plan, but does not make choices between multiple possible plans. That is to say, when a sub-goal has been identified, it is added to the plan. This method of restricting the search space by choosing subgoals could exhibit suboptimal behavior because it commits itself to a bad subgoal and has no way of recovering from this choice. On the other hand, search algorithms such as A* search, while they may be sequential instead of hierarchical, do not commit to one plan and can switch focus between multiple possible plans. This allows room for error while still allowing the algorithm to find shortest paths.

Correctness: To the best of my knowledge, yes.

Clarity: For the most part, the paper is well written. However, there are some key details about the algorithm that are not mentioned. - Line 218: "We train a neural network estimator for the expected cost via supervised learning by randomly sampling two observations from a training trajectory and evaluating the true cost on the connecting trajectory segment C(ot; : : : ; ot0 ) to obtain the target value." How is the true cost determined? By true cost I am assuming this is the cost of a shortest path. Since you use sub-optimal planners to obtain trajectories, then the cost from the planner cannot be considered the true cost. - Line 272 says the agent is tasked to reach the goal on the shortest path. Is this reflected in the success rate? If not, I am not sure why this is being mentioned. - What is the value of M in algorithm 1 in your experiments? - Given a plan in latent space, how does the algorithm produce a policy?

Relation to Prior Work: Line 38 "The above approaches would attempt to model all possible routes starting at home and then search for those that ended up at the airport." This is not true. For example, Hafner et. al, 2019 does not model all possible routes. One major difference between this work and previous work is that this work requires examples of successful trajectories. Other previous work mentioned by the authors does not have this requirement. I think this should be mentioned in the paper. Furthermore, the reasons why the author's algorithm may be preferable should also be discussed. For example, is it able to learn with fewer real-world samples?

Reproducibility: No

Additional Feedback: What is the difference in cost between the sub-optimal planner and this visual planner? Does the visual planner actually learn to find shorter paths than the demonstrations? I only see results for visual prediction for the pick and place problem. Is your method able to solve this problem in the planning and control setting? If not, why not? Minor comments: Citations in supplementary material do not match. I.e. Kavarki et al. [30]


Review 4

Summary and Contributions: In this paper, the authors propose a method for long term planning by breaking it down hierarchically: finding a good point between the beginning and end, then continuing to find points between those until short-term planning algorithms can reach from one point until the next. The authors design a system that can perform this goal-conditioned planning from image data, predicting a path given only the initial and end state images. They test this system across four different datasets, including robot pick and place, human poses, and navigation through rooms, and find that their algorithm outperforms alternative video interpolations, and that the hierarchical planning is particularly important over longer planning horizons.

Strengths: This work is a very interesting approach that can produce arbitrary subgoals for planning in a continuous space. It is theoretically well motivated, and the experiments were well run and evaluated according to standard metrics. This work approaches a problem that is of interest to the NeurIPS community -- long term planning -- and offers a novel way of approaching it.

Weaknesses: One question I was left with after reading this paper is how the model decides when to stop producing heirarchical subgoals. Is there always a set of T observations for each dataset, and therefore the number of nodes in the tree is fixed to T? Or can this be set adaptively so that longer horizon plans get broken down further than shorter horizon plans? In addition, this model seems particularly dependent on the function that predicts S_{T/2} | S_1, S_T. It appears that this requires learning from a fixed set of actions or (for planning) a fixed geometry. Would this function generalize at all? Or would the model require retraining for any new scenario it was required to plan in? How data efficient would this training be?

Correctness: Yes -- the claims made in this paper appear supported by the data

Clarity: Yes, the method and experiments were clearly described

Relation to Prior Work: The authors do describe how this method relates to prior planning algorithms, and how this approach allows hierarchical planning in continuous, image-based domains

Reproducibility: Yes

Additional Feedback:

[Author Response · NeurIPS 2020]

We thank the reviewers for valuable and insightful feedback. The reviewers note that the method is novel, interesting, and relevant to the field. To address the reviewers' concerns about the training data, we provide additional experiments with completely random data collection policy and find that our method improves performance over the data collection policy. We also perform preliminary experiments with robotic manipulation. We will update the manuscript with these experiments, as well as other suggestions, as detailed below.

**R4**: *"Requires examples of successful trajectories", Can the method "find shorter paths than the demonstrations?"*
To reach a particular goal, our method requires training trajectories that reach goals from the same distribution. Note that we still test on *unseen* goals. This is a common requirement for visual planning and control (Ebert'18 [10], Pathak'18 [44]). These trajectories are collected with a suitable exploration policy that need not be optimal but should cover a wide enough trajectory distribution. To test whether our method can work with very suboptimal training data, we conducted a new experiment with completely random exploration data, and observe that our method still successfully solves navigation tasks in the 9-room environment (see Fig 1). This leads us to believe that

Figure 1: **Left**: random exploration data. **Right**: execution of our method trained on random data.

the proposed method is scalable even to situations where no good planners exist that can be used for data collection. We will include a full evaluation of training with random action data in the final version. In Tab. 1, we compare the average trajectory length of training data and our method on both, the dataset from the original submission and the random action data. We find that planning with our method leads to substantially shorter trajectories.

**R3**: *What are the differences with Sub-goal trees [26]?*
[26] employs a stochastic dynamic programming approach for learning to predict trajectories with low cost using a hierarchical predictive model. In contrast, we employ a sampling-based planning approach that hierarchically optimizes the latent variables of our stochastic prediction model *at decision time*. Such

Table 1: Average Trajectory Length. Planning with GCP finds shorter paths than the training distribution.

|  | ORIGINAL DATA | RANDOM DATA |
|---|---|---|
| TRAINING DATA | 31.4 | 62.6 |
| GCP-TREE (OURS) | **20.7** | **42.6** |

decision-time planning allows for greater flexibility, e.g. by changing the cost function post hoc, after training the model. Crucially, the use of *latent variables* allows our model to scale to modeling image sequences and we demonstrate its applicability to long-horizon *visual* control tasks from raw pixel inputs, while [26] only apply their method to low-dimensional state-based tasks. Finally, we propose an adaptive binding scheme for non-balanced subgoal splits that can discover bottleneck states.

**R3, R4**: *Application to new environment / Pick & Place task*
We have now performed preliminary evaluations of GCP-tree in a state-based robotic pick & place environment. Our approach performs long-horizon object manipulations like lifting blocks over a barrier and stacking them (see Fig 2). We will add a full quantitative evaluation with comparisons in the final paper.

**R1**: *"in larger problems [...] optimisation over z infeasible."*
Planning over latent states has similar properties to planning over images, but is more scalable as the latent states are compact. We show that optimizing over z substantially improves the plans over the training data (see Fig. 1, Tab. 1, response the R4 on top). The optimization is indeed harder with longer sequences, but our goal-conditioned prediction and hierarchical planning enable us to optimize well even where prior work fails (e.g. over 200 steps in Fig. 1).

**Time**

Figure 2: Executions of GCP-Tree on a pick & place task with wall separator (subsampled for visibility).

**R4**: *"The algorithm [...] commits itself to a bad subgoal and has no way of recovering from this choice."*
It is quite possible to maintain multiple potential waypoints in parallel, analogously to a beam search. We found this to not be necessary for our tasks, and our method attains substantially better results than prior methods without a beam search, but we will discuss this as a promising topic for future work.

**R4**: *"the agent is tasked to reach the goal on the shortest path. Is this reflected in the success rate?"*
No, but it is reflected in the trajectory cost, which we also report in Table 4 (see L283).

**R3**: *"How does the adaptive binding affect the performance"*
Adaptive binding usually performs comparable or slightly worse due to harder optimization. We expect the benefits of adaptive binding to become more clear with better optimization or where semantic bottlenecks are important.

**R5**: *"Is there always a set of T observations for each dataset"*
Our datasets contain variable-length sequences. In order to determine where to stop hierarchical generation, we use a learned termination classifier at each node. We will add this explanation to the supplement.

[Meta-Review · NeurIPS 2020]

This paper proposes a method for hierarchical, long-horizon planning, by finding good intermediate points between the start state and goal state, then doing the same for the subdivided plans, decomposing the plan into smaller pieces. This goal-conditioned planning is done from image observations, predicting a path given only the initial and end state images. The algorithm outperforms alternative methods based on video interpolation, on scenarios that involve both image-based navigation and robot pick and place. I think reviewers were initially divided on the merits of this paper and what it contributes compared to past work, such as Hindsight Experience Replay (HER) and Searching on the Replay Buffer (SORB), mostly being concerned about the latter. After the rebuttal they converged on the fact that the idea of imagining intermediate goals is useful and concerns about overlap were clarified by the rebuttal. I think the score of 9 is too generous for the paper, given that similar ideas have appeared in the hierarchical generative models literature (even in the video prediction literature) a number of times (e.g. https://zswang666.github.io/P2PVG-Project-Page/, which is not cited but should be). That said, the approach is very promising, and I recommend it be accepted as a poster at the conference.